# Which outcomes are important to patients and families who have experienced paediatric acute respiratory illness? Findings from a mixed methods sequential exploratory study

Michele P Dyson,[1] Kassi Shave,[1] Allison Gates,[1] Ricardo M Fernandes,[2] Shannon D Scott,[3] Lisa Hartling[1]

[1]Alberta Research Centre for Health Evidence and the Department of Pediatrics, University of Alberta, Edmonton, Alberta, Canada
[2]Department of Pediatrics, Hospital de Santa Maria, Lisbon, Portugal
[3]Faculty of Nursing, University of Alberta, Edmonton, Alberta, Canada

**Correspondence to**
Dr Michele P Dyson;
mdyson@ualberta.ca

## ABSTRACT

**Objectives** To identify the outcome priorities of parents of children who had experienced an acute respiratory infection (ARI).

**Design** This was a two-phase, mixed methods study with a sequential exploratory design. We used a cross-sectional quantitative web-based survey to elicit parents' priorities for paediatric ARI. We then used a discussion moderated via Facebook to elucidate richer descriptions of parents' priorities.

**Setting** Survey and discussion data were collected via the internet.

**Participants** 110 parents (90% women, median age 35 years, 92.7% urban dwelling, 94.5% with a postsecondary education) with a child who had experienced an ARI responded to the survey. Four parents participated in the Facebook discussion.

**Primary and secondary outcome measures** The primary outcome was parents' rankings of outcomes related to paediatric ARI. The secondary outcomes were the alignment of parent-reported important outcomes with those commonly reported in Cochrane systematic reviews (SRs).

**Results** Commonly reported ARIs included croup (44.5%), wheezing (43.6%) and influenza (38.2%). Parents ranked major complications, illness symptoms and length of stay as the most important outcome categories. With respect to specific outcomes, severe complications, major side effects, doctor's assessment, relapse, oxygen supplementation and results from laboratory measures were reported as most important (75th–99th percentile). Taking time off work, mild complications, interference with daily activities, treatment costs, absenteeism, follow-up visits and other costs were deemed minimally important (<25th percentile). In 35 Cochrane SRs, 29 unique outcomes were reported. Although participants' priorities sometimes aligned with outcomes frequently reported in the literature, this was not always true. Additional priorities from the survey (n=50) and Facebook discussions (n=4) included healthcare access, interacting with healthcare providers, education, impact on daily activities and child well-being.

**Conclusions** In the context of paediatric ARI, parents' priorities did not always align with commonly researched outcomes. Appealing and efficient strategies to engage patients and parents in research should be developed.

## Strengths and limitations of this study

► The identification of patient-important outcomes is a necessary precursor to the conduct of research that is relevant to themselves and their families, though engaging patients in research is challenging.

► We used web-based tools and social media platforms to recruit and engage patients and identify patient-important outcomes for paediatric acute respiratory infections.

► Though nearly three-quarters of online adults use social media, engaging in health research online may appeal only to certain subpopulations, so the findings may not be generalisable.

## INTRODUCTION

The determination of outcomes that matter to patients is foundational to the conduct of research that is relevant to them and their families. With an increasing emphasis on patient-centredness in clinical research, numerous organisations and strategies have been established with this as their mandate (eg, Canada's Strategy for Patient-Oriented Research,[1] the Patient-Centered Outcomes Research Institute in the USA[2] and INVOLVE in the UK).[3] Involving patients in the research process will ensure that funded investigations use questions, outcomes and interventions that are aligned with their needs and priorities.[4–6]

There are many complexities involved in selecting outcomes, and little published guidance for investigators exists.[7] There is significant heterogeneity in the outcomes measured and reported in studies of specific diseases, which may in part occur due to uncertainty around which outcomes are patient-important.[7] The development of core outcome sets, in which a minimum group

of agreed-upon outcomes is measured and reported on across clinical research in a specific condition, has been proposed as a solution to these issues.[8] Sinha *et al*[7] identified 13 groups formed to develop core outcome sets for paediatric clinical trials, including the Core Outcome Measures in Effectiveness Trials (COMET) and Outcome Measures in Rheumatology (OMERACT). COMET was launched in 2010 to bring together individuals interested in developing core outcome sets, and to collate outcome sets and relevant resources.[8] Established in 1992, OMERACT is a consensus initiative that has developed a number of widely used core outcome sets for rheumatological conditions, with patients actively involved in the process since 2002.[9 10]

One criticism of commonly used methods to develop core outcome sets is that they do not include a systematic survey of stakeholders.[9] Social media represents a medium where patients and their caregivers increasingly interact online,[11 12] providing an opportune channel for engagement in the development of core outcomes. Nearly three-quarters (74%) of online adults use social media, with Facebook continuing to be the most popular social media site, and multiplatform use increasing in prevalence.[13 14] Despite the global pervasiveness of social media, its use for engaging patients and/or caregivers in the outcome selection process has not extensively been explored.

We conducted a two-phase, sequential exploratory mixed methods study using social media to recruit and engage the parents of children with an acute respiratory infection (ARI) to elucidate patient-important outcomes. ARIs are common among children, represent a significant source of morbidity and are one of the leading causes of illness, emergency department visits and hospitalisation.[15 16] There is a scarcity of research on patient perspectives in this area.[7] Using social media, we aimed to: (1) recruit and survey parents to identify their priorities for ARI outcomes as compared with those commonly reported in the literature, and (2) engage parents in discussions to elucidate the rationale for their priorities, as identified in the quantitative survey.

## METHODS

This mixed methods study used an explanatory sequential design,[17] and involved two phases: (1) a quantitative survey to determine parent priorities for ARI outcomes; (2) a qualitative follow-up in which we sought elaboration on parents' priorities. A process evaluation of our social media strategy is reported elsewhere.[18]

### Institutional ethics approval

Prior to beginning the study, we sought and received ethical approval for both the quantitative and qualitative components. Parents and/or caregivers were eligible to participate if they had a child aged 0–17 years who had experienced one or more episodes of acute asthma, bronchiolitis, croup, influenza, strep throat/ tonsillitis, pneumonia, sinusitis and/or wheezing. Participant consent was implied through overt action by completing the survey or publishing public responses online. Prior to participation, we provided interested potential participants with an information letter that described the study and explained consent via overt action. Participants were free to withdraw, end or modify their participation in the study at any time without consequence, and we retained any data collected only with their permission.

### Phase I: quantitative survey
#### Development and pretesting

The first draft of the survey was informed by previous research on the outcomes that are important to clinicians and families of children with asthma[19] and by outcomes frequently reported in the literature. To determine the frequency of outcomes reported in the literature, we identified systematic reviews (SRs) published up to 2013 from the Cochrane Database of Systematic Reviews (n=35) and their included studies. We grouped frequently reported outcomes and those previously identified as important[7 19 20] into seven broad categories: disease activity, disease complications; adverse effects of therapy; functional status; social and family outcomes, including quality of life; long-term effects of interventions and resource utilisation. We sent a draft of our survey to a group of 10 clinicians/researchers and 8 parents who reviewed it independently and provided feedback on its content and comprehensibility. We analysed the written feedback qualitatively. Based on the content analysis, our research team revised the original categories, finally deciding on 10 categories of outcomes for the survey. These included outcomes that were identified as important by clinicians and parents that were missing from the draft. The 10 categories on the final version of the survey included: major complications; symptoms; length of stay in the emergency department or hospital; needing to see a doctor; returns visits to a doctor or the hospital; reactions to medications; medical test results; maintenance of day-to-day activities; minor complications and cost of illness. The survey is available as a supplementary file (online supplementary file 1).

We ensured survey accessibility across different operating systems, including touch screen (eg, tablets, smartphones) or keyboard (eg, desktop computers) technology. We wrote the materials at a sixth-grade reading level and conducted pilot testing with 8 parents and 10 researchers/clinicians to ensure that the language and flow of questions were appropriate. We ascertained the reading level of our materials via the readability statistics provided in Microsoft Office Word's proofing options. With the readability statistics option turned on, Word returns the Flesch-Kincaid Grade Level of the document or the highlighted text following proofing for spelling and grammar.

## Recruitment

We created an online and social media presence via a study website (Outcomes in Child Health; www.outch-study.com), a Facebook page (OUTCH) and a Twitter account (@OUTCH_Study). We used snowball sampling[21] to recruit parents. First, we focused on identifying and engaging recruitment targets with the potential for a high yield of participants. We then expanded our scope through referrals and diffusion via social media. Tactica Interactive (https://tactica.ca), a digital media enterprise, was hired to broaden our sampling frame via a Facebook advertising strategy.

We collaborated with organisations interested in ARI and patient engagement to advertise our research via websites and other channels: The Alberta Centre for Child, Family & Community Research (now known as PolicyWise for Children and Families; a provincial organisation linking government, academia and the community in a focus on evidence-informed policy and practice),[22] TRanslating Emergency Knowledge for Kids (a national network of researchers and clinicians invested in improving paediatric emergency care),[23] the Cochrane Consumer Network (an international network of healthcare consumers with an interest in evidence-based medicine)[24] and the Stollery Family Centered Care Network (a local children's hospital-based network of patients and families that provide input into patient care).[25] We also engaged an online (Facebook and Twitter) parenting community, Mommy Connections, that regularly promoted the study through their networks.

## Data collection

The quantitative survey was administered by Nooro Online Research (https://nooro.com/index.html) for 14 weeks from December 2013 to March 2014. Links to the survey were provided through the study website, Facebook and Twitter accounts and were completed anonymously. The only identifying information was an optional email address for entry into a prize draw.

The survey included a combination of open-ended and closed-ended questions to determine the relative importance of outcomes currently measured in trials and SRs of ARI in children. The outcome categories were presented and parents were asked to identify their top five priorities from the list. Then, individual outcomes were presented and parents were asked to indicate their importance using a digital sliding scale, conceptually similar to a visual analogue scale. The sliding scale asked parents to rank the importance of each outcome by providing it with a score ranging from 1 (not important (/concerning) at all) to 100 (extremely important (/concerning)). Parents were also asked to indicate additional items that were considered important to patients and their families, but may not have been addressed in the literature. The survey platform was also used to collect demographic data.

## Phase II: qualitative follow-up

In the second phase, we conducted an internet-based, descriptive qualitative study[26] to interact with parents and elicit elaboration on their perceptions of the importance of ARI outcomes. This study occurred across an 8-week period from January to March 2016 during which parents were engaged in discussion through an open online focus group hosted on the study Facebook page.

## Recruitment

To recruit parents, we used a snowball sampling technique.[21] We first asked potentially high yield sources of participants to promote our study, including local and national online parenting communities (n=16), children's hospitals and associated foundations (n=14) and patient groups (n=3) with access to a large consumer audience. We also asked individuals and organisations within our existing networks to promote the study. Recruitment occurred throughout the study period, and was almost exclusively carried out through Facebook. Some organisations or individuals promoted the study on Twitter or a blog, but all links posted drove user traffic back to the study Facebook page.

One variable of interest in this study was the reach of social media as a recruitment strategy (reported elsewhere[18]); therefore, the sample size was an outcome, rather than a predefined condition. To accommodate this, we did not define the number of participants a priori, instead allowing the detail that emerged from our data collection to guide the extent of recruitment. However, we did aim to recruit a sample size as guided by the principles of data saturation, in which data would be collected until no new themes emerged.[27]

## Data collection

Throughout the study period, each week had a discussion theme modelled after the structure of the survey, and posts were published daily covering varying aspects of this theme. Three different types of posts were published: promotional posts prior to the study launch; parent-friendly content about ARIs and discussion questions where parents were encouraged to share their thoughts and experiences. The online focus group was moderated by two members of the study team (MPD, KS) and all posts by participants were followed up promptly with a response. An interview guide (online supplementary file 2) was developed to guide weekly topics and for reference during discussion moderation.

## Data analysis
### Quantitative analysis

Survey data were analysed using SPSS (V.22.0, IBM) and described using descriptive statistics (mean±SD; rank order). To determine the rank order of the outcomes most important to parents, we allocated each outcome 10 points when it was chosen as a top concern, and 8, 6, 4 and 2 points if it was chosen as the second, third, fourth or fifth most important concern, respectively. After tallying

the points for each of the 10 outcomes, we ordered these from largest to smallest to develop the ranked priority list.

To compare the published literature to the patient-important outcomes identified by parents, we collated a list of all of the outcomes, and calculated the number of SRs in which each outcome had been reported. We then grouped all of the outcomes into percentile ranges based on the number of SRs in which each was reported ('frequently reported': 75th–99th percentile; 'moderately reported': 25th–74.9th percentile and 'infrequently reported': <25th percentile). Similarly, we calculated the mean score given to each outcome by parents on the digital sliding scale (ie, scores from 1 to 100), and grouped the outcomes into percentile ranges ('most important': 75th–99th percentile; 'moderately important': 25th–74.9th percentile and 'least important': <25th percentile). All comparisons were strictly descriptive. Parents who had healthcare insurance were self-identified in the survey and excluded from all analyses related to healthcare costs as these may not hold relevance.

### Qualitative analysis

Content posted by participants during the focus groups was extracted verbatim to form transcripts. These, along with the open-ended responses to the survey, were imported into NVivo V.10 (QSR International) data management software. Data were analysed inductively for themes. Two investigators (MPD, KS) participated in coding, following a three-stage process: (1) reading through the data, making notes on themes and significance that were then compiled into a preliminary version of the codebook; (2) rereading the data and coding using the concepts identified in the first phase and (3) refining and applying the codes to the text on a third review.[28] Coders met to discuss progress and reach consensus on differing interpretations. Data collection and analysis occurred concurrently, following an iterative process to monitor progress and allow for follow-up on ideas as they emerged.

## RESULTS

### Demographic characteristics

A total of 110 people responded to the survey (table 1). The survey website received 5207 visits, a view rate (ie, the ratio of unique survey visitors/unique site visitors) of 3.9% (205/5207) and a completion rate (ie, the ratio of unique visitors who completed the survey/users who agreed to participate) of 53.7% (110/205). A detailed account of the traffic to the survey and usability are available in our published process evaluation.[18]

Ninety per cent (n=99) of respondents were women. The median age was 35 years, and 88% (n=97) of respondents had received a college/university or postgraduate education. Most survey respondents were married (n=98; 89%), urban dwelling (n=102; 93%) and resided in Canada (n=77; 70%). More than half of respondents reported an annual household income of >US$90 000 (n=69; 63%).

Respondents were predominantly parents (n=106; 96%), and had a median of two children in the home (range 0–4). The respondents' children most commonly experienced croup (n=49; 45%), wheezing (n=48; 44%) and influenza (n=42; 38%). The most concerning ARIs were croup (n=20; 26%), pneumonia (n=16; 21%) and asthma (n=15; 19%). The median year that the ARI occurred was 2012 (range 1994–2013), when the child was 1 year old (range <1 month to 10 years old). Most children did not have a chronic illness (n=90; 82%) and did not experience a hospital admission due to this ARI (n=84; 76%).

### Quantitative outcome rankings

The overall ranking of categorised outcomes is shown in table 2. On average, parents ranked major complications from the child's illness (eg, long-term disability), illness symptoms (eg, coughing, fever, sore throat) and length of stay in the emergency department or hospital as the most important outcome categories. Of least importance were the costs of their child's illness (eg, medicine or child care), minor complications from the child's illness (eg, cough or rash) and maintenance of day-to-day activities. The overall ranking of individual outcomes revealed that parents were most concerned about severe complications (mean score on a scale from 1 to 100: 94.5), major side effects (86.7) and their doctor's assessment (83.9). Other costs (eg, child care, parking, lost income) (31.3), scheduled follow-up visits (38.1) and school/daycare absenteeism (40.4) received the lowest mean scores. When parents were grouped according to ARI their child had experienced and their rankings of the importance of outcomes were compared, Spearman correlation coefficients revealed strong agreement, indicating that perceptions of importance were consistent across conditions (table 3).

A comparison of the outcomes reported in the literature to those reported as important by parents is shown in table 4. We extracted 221 outcomes from the 35 Cochrane SRs; the same outcomes were often reported in more than one SR. Out of the full list of reported outcomes, we isolated 29 individual outcomes, each of which was reported in 1 to 26 of the SRs. Adverse events were the most frequently measured outcome (f=26; 11.8%) in our sample of Cochrane SRs, and similarly, severe complications (score: 94.5/100) and major side effects (86.7) were ranked as the most important (75th–99th percentile) to parents. Likewise, parents ranked returning to school/work and the cost of treatment as least important (40.4 and 44, respectively; <25th percentile), and these outcomes were infrequently measured in Cochrane SRs (f=2; <25th percentile).

There were many discrepancies between the outcomes measured in Cochrane SRs and the outcomes parents ranked as important. Relapse and the need for oxygen supplementation were ranked among the most important outcomes by parents (81.8 and 81.6, respectively; 75th–99th percentile), but were measured moderately frequently (f=8 and f=5, respectively; 25th–74.9th

| Table 1 Survey participant demographics (n=110) | |
|---|---|
| **Characteristic** | **n (%)** |
| Gender | |
| Female | 99 (90.0) |
| Male | 11 (10.0) |
| Age (years) (median (range)) | 35 (18–67) |
| Highest level of education | |
| Some high school | 1 (0.9) |
| High school graduate | 5 (4.5) |
| Some college/university | 7 (6.4) |
| College/university graduate | 50 (45.5) |
| Postgraduate education/degree | 47 (42.7) |
| Marital status | |
| Single | 5 (4.5) |
| Married/Common-law | 98 (89.1) |
| Separated/divorced/widowed | 7 (6.4) |
| Annual household income (US$) | |
| <30 000 | 5 (4.5) |
| 30–49 999 | 7 (6.4) |
| 50–69 999 | 15 (13.6) |
| 70–89 999 | 14 (12.7) |
| >90 000 | 69 (62.7) |
| Country of residence | |
| Australia | 2 (1.8) |
| Canada | 77 (70.0) |
| England | 8 (7.3) |
| India | 2 (1.8) |
| Portugal | 2 (1.8) |
| USA | 19 (17.3) |
| Type of community | |
| Urban (≥10 000) | 102 (92.7) |
| Rural (<10 000) | 7 (6.4) |
| Missing | 1 (0.9) |
| Number of children in home (median (range)) | 2 (0–4) |
| Relationship to child | |
| Parent | 106 (96.4) |
| Step-parent | 0 (0) |
| Grandparent | 4 (3.6) |
| Other | 2 (1.8) |
| Type of ARI | |
| Bronchiolitis | 29 (26.4) |
| Croup | 49 (44.5) |
| Strep throat/tonsillitis | 36 (32.7) |
| Wheezing | 48 (43.6) |
| Influenza | 42 (38.2) |
| Pneumonia | 24 (21.8) |

Continued

| Table 1 Continued | |
|---|---|
| **Characteristic** | **n (%)** |
| Asthma | 29 (26.4) |
| Other | 23 (20.9) |
| Most concerning ARI | |
| Bronchiolitis | 10 (13) |
| Croup | 20 (26) |
| Strep throat/tonsillitis | 3 (4) |
| Wheezing | 6 (8) |
| Influenza | 2 (3) |
| Pneumonia | 16 (21) |
| Asthma | 15 (19) |
| Other | 6 (8) |
| Year of ARI (median (range)) | 2012 (1994–2013) |
| Child age at time of ARI (median (range)) | 1 year (<1 month to 10 years) |
| Hospital admissions due to ARI | |
| Yes | 21 (19.1) |
| No | 84 (76.4) |
| Missing | 5 (4.5) |
| Chronic illness | |
| Yes | 20 (18) |
| No | 90 (82) |

ARI, acute respiratory infection.

percentile) in Cochrane SRs. Similarly, the results from laboratory measures were ranked highly by parents (81.4; 75th–99th percentile), but were infrequently reported in Cochrane SRs (f=4; 25th–74.9th percentile). The need for medication was one of the most frequently reported

| Table 2 Overall ranking of categorised items (n=110) | |
|---|---|
| **Rank order*** | **Category** |
| 1 | Major complications from child's illness (eg, long-term disability) |
| 2 | Illness symptoms (eg, coughing, fever, sore throat) |
| 3 | Length of stay in the emergency department or hospital |
| 4 | Child needing to see a doctor |
| 5 | Return visits to the doctor or hospital |
| 6 | Child's reaction to his or her medicine (eg, side effects) |
| 7 | Child's medical test results |
| 8 | Maintenance of day-to-day activities |
| 9 | Minor complications from child's illness (eg, cough or rash) |
| 10 | Costs of child's illness (eg, medicine or child care) |

*Ordered from most to least important to parents.

**Table 3** Agreement between acute respiratory infection type and parent ranking of categorised outcomes (n=110)

| Type of acute respiratory infection | Spearman correlation coefficient* |
|---|---|
| Bronchiolitis | 0.94 |
| Croup | 0.75 |
| Strep throat/Tonsillitis | 0.66 |
| Sinusitis | 0.87 |
| Wheezing | 0.76 |
| Influenza | 0.66 |
| Pneumonia | 0.85 |
| Asthma | 0.67 |
| Other | 0.50 |

*0–0.2: poor/slight agreement; 0.2–0.4: fair agreement; 0.4–0.6: moderate agreement; 0.6–0.8: substantial agreement; 0.8–1: near-perfect agreement.

outcomes in Cochrane reviews (f=13; 75th–99th percentile), but was ranked as moderately important by parents (55.1; 25th–74.9th percentile). Admission rate was the second most frequently measured outcome in Cochrane SRs (f=19; 75th–99th percentile), yet was also ranked less favourably among parents (76.9; 25th–74.9th percentile). While the doctor's assessment of how the child is doing was ranked as the third most important outcome (83.9; 75th–99th percentile) by parents, only clinical scores/symptom scores, one of three corresponding components of this outcome (clinical scores/symptom scores; patient improvement; observed response to treatment) was measured frequently (f=15; 75th–99th percentile) in Cochrane SRs. Patient improvement was measured moderately frequently (f=6; 25th–74.9th percentile), and observed response to treatment was measured the least frequently of any outcome included in this study (f=1; <25th percentile).

## Qualitative synthesis

A total of 50 respondents provided qualitative responses on the survey. Four participants contributed to the discussion on the Facebook page. The thematic analysis revealed five main analytical themes relating to parents' priorities and concerns when their child had an ARI: 'accessing healthcare'; 'interacting with healthcare providers'; 'illness education'; 'impact of illness on daily activities' and 'child well-being'.

## Accessing healthcare

Though not expressed by all parents, having timely access to healthcare for their child was a primary concern for many. Being able to get the medical advice they needed, without encountering substantial waiting times was important. Parents expressed concern about recognising the signs and symptoms of their child's disease process. Determining whether their child's symptoms were serious or minor was considered challenging. Parents lacked confidence in deciding when to seek medical attention, particularly when the child frequently experienced breathing problems. Overall, parents shared wanting to avoid making unnecessary healthcare visits, and expressed the importance of receiving helpful tips from healthcare providers for managing acute symptoms at home.

## Interacting with healthcare providers

When parents accessed healthcare for their child, they desired to be taken seriously by healthcare providers. Parents described concern about their child's healthcare provider being dismissive or uninterested in their chief complaints about their child's health. Strong communication with healthcare providers was widely valued by parents. Parents expressed wanting to feel heard, and to have a medical team that was both helpful and thorough in explaining their findings in a way that was direct and understandable.

## Illness education

Receiving education about their child's illness was widely regarded as important. Parents described requiring a complete understanding of what to expect, when to seek medical attention and what the recovery time and process would be like. Information about the long-term impact of their child's health condition, and what effect recurrences or exacerbations might have, was regarded as of specific importance. When appropriate, parents regarded being presented with a variety of treatment options as critical. Parents expressed wanting to be involved in their child's recovery, and regarded education about their child's illness as essential to taking an active role.

## Impact of illness on daily activities

Parents described concern around interruption of their work schedules and sleeping routines. Being able to stay at home or in hospital with their child when they were unwell was important to parents, while not always possible when balancing financial and care responsibilities for other siblings. Disruption of sleep routines for parents and siblings was also described as concerning, particularly among parents of children with uncontrolled cough. Potential spread of their child's ARI between siblings and to the parents themselves was noted as a primary concern, and highly disruptive to daily activities.

## Child well-being

Parents shared their concern for their child's psychological well-being when they had an ARI. Concern about how their child was coping when they were unwell, particularly for those children with recurring or chronic ARI, was emphasised. Parents also expressed concern about how others treated their child when they were sick. One parent described concern over their child being treated like an 'invalid' when experiencing an acute asthma exacerbation.

**Table 4** Parent ranking of individual outcomes compared with frequency of measurement in Cochrane systematic reviews

| Outcomes reported in Cochrane systematic reviews (n=35) | Frequency of reporting | Parent ranking of importance of outcomes (n=110) | Mean score±SD (/100) |
|---|---|---|---|
| **75th–99th percentile** | | | |
| Adverse events (local and systemic) | 26 | Severe complications | 94.5±14.5 |
| Admission rate (hospital, ED, ICU) | 19 | Major side effect | 86.7±18.9 |
| Physical signs | 18 | Doctor's assessment | 83.9±19.6 |
| Clinical measures | 17 | Relapse | 81.8±18.4 |
| Clinical scores/symptom scores | 15 | Oxygen supplementation | 81.6±21.6 |
| Length of stay/time to discharge (hospital, ED, ICU) | 15 | Results from lab measures | 81.4±19.4 |
| Need for medication | 13 | | |
| **25th–74.9th percentile** | | | |
| Severity of symptoms | 12 | Length of hospital stay | 78.6±22.1 |
| Duration of symptoms | 10 | Trip to emergency department | 77.6±23.9 |
| Complications | 9 | Time to recovery | 77.4±17.8 |
| Rates of relapse | 8 | Hospital admission | 76.9±23.7 |
| Clinical treatment failure | 7 | Return healthcare visit | 76.9±24.0 |
| Patient improvement | 6 | Not eating/drinking well | 65.9±22.8 |
| Time to resolution of illness/time to recovery | 6 | Lack of sleep | 63.2±20.0 |
| Duration of oxygen supplementation | 5 | Length of stay in emergency department | 62.4±26.1 |
| Mortality | 5 | Minor side effect | 55.1±24.8 |
| Laboratory measures | 4 | Prescription for medication | 55.1±25.6 |
| Readmission | 4 | Appointment with GP/paediatrician | 48.4±27.8 |
| | | Arranging child care | 48.3±30.1 |
| **<25th percentile** | | | |
| Clinical cure | 3 | Taking time off work | 47.2±28.4 |
| Compliance and tolerance | 3 | Mild complications | 46.3±23.3 |
| Quality of life/patients' well-being | 3 | Interference with daily activities | 44.6±23.5 |
| Return healthcare visits | 3 | Treatment costs | 44.0±32.5 |
| Return to school/work | 2 | School/daycare absenteeism | 40.4±26.5 |
| GP visits | 2 | Scheduled follow-up visits | 38.1±24.8 |
| Treatment cost | 2 | Other costs | 31.3±31.3 |
| Adverse events that necessitated discontinuation of treatment | 1 | | |
| Sleep disturbance | 1 | | |
| Parental perception of child's status | 1 | | |
| Observed response to treatment | 1 | | |

ED, emergency department; GP, general practitioner; ICU, intensive care unit.

## DISCUSSION

Knowledge of patients' and their families' priorities is essential to guide the conduct of research that is relevant to themselves as well as to clinicians and policy-makers.[19] Using social media, we engaged >100 parents over 14 weeks in a survey to elucidate the outcomes that they deemed most important with regard to paediatric ARIs. Parents' most important concerns included clinical outcomes like major complications, symptoms and length of stay in the emergency department or hospital. Psychosocial outcomes, and the ability of the family to cope during a child's illness, were also important. Not surprisingly, parents who participated in the focus groups were also concerned about process measures, like wait times, communication with healthcare providers and managing their child's care at home. Although parents

did not explicitly make the link, research has indicated that certain care processes, for example, patient-centredness, may contribute to better health outcomes.[29] For example, family-centred care is associated with improved clinical outcomes for children and greater satisfaction with care.[30] The approach is characterised by honest communication between families and healthcare providers; policies and procedures that are tailored to the needs of families and children; ensuring support for families and children and empowering them to participate in care decisions.[30]

The maintenance of day-to-day activities was of relatively low importance to parents, seemingly contradicting the highly ranked importance of major complications and long-term disability. A previous study of asthma outcomes found that parents were more concerned about the long-term compared with short-term beneficial and harmful effects of therapy.[19] Because we did not quantify the temporality of the outcomes on the survey or in the discussions, we can only presume that parents interpreted the maintenance of day-to-day activities as a short-term outcome resulting from relatively minor illness. Although major complications can result in long-term or permanent changes to daily routines, they could also be potentially life threatening. When presented with the possibility of serious complications that could limit long-term quality of life, the maintenance of one's daily routine may seem relatively unimportant.

Health, though difficult to define, encompasses not only an individual's physical condition, but also their emotional and psychological well-being.[31] Our survey of 35 Cochrane SRs revealed that a diverse array of health outcomes are being measured and reported, many of which are not aligned with those that are important to parents. These findings reinforce the growing recognition that insufficient consideration is being paid to the selection of outcomes within clinical trials.[32] Specifically, the need for core outcome sets,[33] and especially ones that incorporate patient-reported and patient-centred outcomes,[34] has garnered increased attention in recent years. More consistent reporting of outcomes for paediatric ARI will be necessary to facilitate evidence synthesis,[32 34] to enhance trustworthiness by reducing the risk of reporting bias[35] and to reduce research waste.[36]

Good practice in clinical trials includes selecting a primary outcome that measures a clinically relevant and important treatment benefit.[37] Likewise, we found that the bulk of the research in child health focused on biological outcomes, with relatively little attention being paid the psychosocial impact of illness.[20 38 39] Still, there remains room in research for the measurement of outcomes important to patients and their families. Standards for Research in Child Health, founded in 2009, brings together clinical and methodological experts to develop and promote the uptake of evidence-based guidance for child health research.[40] They assert that trialists should measure the effects of interventions more comprehensively; by measuring long-term outcomes and those that are relevant to decision-makers and families, the findings of trials will be of greater value.[20]

Moving toward greater inclusion of patient-important outcomes in paediatric health research is challenged by the fact that children and their parents can be difficult to reach and engage. Given the pervasiveness of social media use via multiple platforms by patients and their caregivers,[41] we postulated that this would provide an opportune medium to learn parents' perspectives. Though we experienced relative success in recruiting parents to complete the survey, qualitative engagement via the Facebook discussion group was more arduous. Moreover, despite moderate success in engaging parents, we did not gather any information from children themselves. Children have the right to participate in matters that affect their own lives,[42] and can provide unique perspectives that cannot be elicited from their caregivers. Nevertheless, children also require protection, and the extent to which minors can understand and express their own healthcare needs remains controversial.[42]

The challenges that we experienced are not unique. A review of studies that addressed the process of outcome selection identified only three studies that involved parents and none that involved children in the identification of paediatric patient-important outcomes.[7] In deciding which outcomes should be measured in paediatric ARI, it will be essential that stakeholders with varied perspectives, including parents, children, researchers, clinicians and decision-makers, convene and reach agreement on research priorities.[32] Suggested approaches like the Delphi technique and nominal group technique provide a means for stakeholders to reach unanimity on important outcomes in child health research.[7 32] These are, however, time-consuming, resource-intensive and are highly burdensome to participants, which may limit recruitment and engagement. Further guidance is required on consensus methods that are efficient and appealing to patients, families and other stakeholders. Methods that are understandable to children will need to be developed if researchers are to uphold the rights of minors to be involved in their own healthcare.[42] Reconciling children's and parents' perspectives, and the extent to which minors should be involved in the consensus process, requires further study.

## Limitations

Our sample of parents and guardians were highly educated, many of whom had family incomes well above the national median, were mostly urban dwelling and mainly Canadian, limiting the generalisability of the findings. For example, participants with lower incomes or those residing in countries without publicly funded healthcare may have placed more importance on the cost of illness. As we did not provide any details to parents as to the temporality of the outcomes on our survey and in our discussions, we were not able to determine whether short-term or long-term complications were more important to parents.

We had great difficulty engaging parents in the qualitative discussion and only elucidated responses from four participants. This seriously limited our ability to make informed inferences with regard to parents' quantitative ranking of the outcomes, leaving these mainly open to interpretation. For this reason, our understanding of the reasoning behind parents' ranking of the outcomes and the content of the emergent qualitative themes are preliminary. Further work is required to develop a more comprehensive understanding of why some ARI-related outcomes are more important to parents than others.

## CONCLUSIONS

The conduct and reporting of research of little relevance to the primary stakeholders represents a significant source of research waste, and appears prevalent in the context of paediatric ARI. The development of core outcome sets that include patient-important outcomes will facilitate evidence synthesis and reduce reporting bias, supporting the utility and trustworthiness of research findings. Future investigations are required to elucidate ways to make engagement in research more efficient and appealing to patients and their families.

**Acknowledgements** The authors would like to thank Sarah Oberik-Olivieri for her graphic design work for this study.

**Contributors** MPD and LH designed the study. MPD oversaw all aspects of the study's implementation. MPD and KS collected, analysed and interpreted the data with input from AG, LH, RMF and SDS. All authors had full access to the data, and can take responsibility for the integrity of the data and the accuracy of the data analysis. KS and AG drafted the manuscript. MPD, LH, RMF and SDS reviewed the manuscript critically for intellectual content. All authors approved the version of the manuscript that was submitted to the journal.

**Funding** This work was supported by KT Canada, grant number CIHR 87776 (SG-1) and Alberta Innovates Health Solutions, grant number 201400561. LH is supported in part by a Canadian Institutes of Health Research New Investigator Award. SDS is a Canada Research Chair (Tier II) for Knowledge Translation in Child Health and is also supported by an Alberta Innovates Health Solutions Population Health Investigator Award.

**Disclaimer** The funders had no role in study design, data collection and analysis, decision to publish or preparation of the manuscript.

**Competing interests** None declared.

**Ethics approval** University of Alberta Research Ethics Board (number Pro00058629).

**Provenance and peer review** Not commissioned; externally peer reviewed.

**Data sharing statement** Anonymised data are available from the corresponding author upon reasonable request.

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
