## [Reviewer comments · BMJ Open]

ARTICLE DETAILS

TITLE (PROVISIONAL)	Which outcomes are important to patients and families who have experienced pediatric acute respiratory illness? Findings from a mixed methods sequential exploratory study
AUTHORS	Dyson, Michele; Shave, Kassi; Fernandes, Ricardo; Scott, Shannon; Hartling, Lisa; Gates, Allison

VERSION 1 – REVIEW

REVIEWER	Philip van der Wees Radboudumc, Netherlands
REVIEW RETURNED	04-Jul-2017

GENERAL COMMENTS	This is an interesting study describing the results of a novel data collection method via social media, of which the process evaluation has been published elsewhere. The presentation of the results in the present study provides new insights in outcomes deemed important by parent of children after acute respiratory infection. He study shows that outcomes typically used in clinical trials are not always deemed relevant to the parents. I have several comments and queries which should be addressed by the authors in clarifying the methodology ad results of the study: 1. Ethics statement: an ethics statement is lacking in the manuscript2. Survey development: The description of the survey development lacks specification necessary fir understanding the process and choices made:a. What process did the authors undergo to reach consensus for the categoriesb. I do not understand the how the transition was made from the initial seven categories to the final ten categories. The content of the seven vs. ten categories seems quite different. Some of the ten categories seem to derive from the in initial list (e.g. from disease complications to major and minor complications), but for other categories (e.g. length of stay, needing to see a doctor) I found it difficult to see the transition.c. What and how many content experts and consumers were involved in the pilot testing of the categories (similar as the survey testing?) and which three outcomes were added?3. Final version of the survey: Please provide more insight in the final version of the survey, including presentation as supplementary file. If I understand it well, the ten categories contained individual underlying outcomes? How many outcomes per category were presented. How long did it take to complete the survey?
--

	4. Outcomes identified: I am confused about the low ranking of 'maintenance of day-to-day' activities. This seems in contract with the high ranking of long-term disability under the category major complications. Long-term disability typically refers to limitation in activities and social participation. Please clarify this (apparent) discrepancy. Could it be that the parents did not fully understand these concepts? Especially, since the parents mentioned impact on daily activities in the qualitative assessment via the Facebook sessions. Or am I mixing limitations of the children vs. their parents in daily activities? 5. Related to the comment above: did the questions and discussions distinguish in short-term vs. long term outcomes? I would argue that parents are concerned about acute symptoms on the short term, while long-term outcomes would be more related to daily activities. 6. Outcomes reported in the literature: I do not understand what 29 'unique' outcomes means, derived from the 221 outcomes identified. Do you mean grouping of the 221 outcomes in 29 clusters of outcomes? 7. Facebook discussions: An interesting finding to me was that the Facebook discussions revealed other themes more related to the process of care (interaction with health care providers, illness education) than outcomes of care. I suggest reflecting on these differences in the Discussion section. 8. Only four parents contributed to the Facebook discussions The authors should even stronger emphasize this limitation. 9. I suggest further reflection on comparing outcomes identified via the parents vs. the literature. Can they really be compared?
--	--

REVIEWER	Mikael Lavigne North York General Hospital, Toronto, Ontario, Canada; University of Toronto, Toronto, Ontario, Canada
REVIEW RETURNED	19-Sep-2017

GENERAL COMMENTS	This is an interesting and well designed study. In table 1, one of the ARI conditions is "Strep throat/tonsillitis" whereas in the Methods section it is described as pharyngitis/tonsillitis so the study authors could consider choosing one of the two for consistency. The categorized items are shown as ranked by the parents in table 2, but there is no description of how the ranking calculation is done - could this be clarified? i.e. it would be useful to know how many parents thought the different categories were important, since there is a difference between a 1st ranked outcome that was ranked by 100/110 parents vs a 10th ranked outcome that might only be ranked by 1/110 parents. In addition, the outcomes selected for the survey are described as being obtained from Cochrane systematic reviews, but it would be helpful to know how these were selected - were all outcomes described in the SRs retained? Did some of the study authors select which outcomes were to be retained in collaboration with the individuals involved in the pilot testing portion?
--

VERSION 1 – AUTHOR RESPONSE

Reviewer 1

1. This is an interesting study describing the results of a novel data collection method via social media, of which the process evaluation has been published elsewhere. The presentation of the results in the present study provides new insights in outcomes deemed important by parent of children after acute respiratory infection. The study shows that outcomes typically used in clinical trials are not always deemed relevant to the parents. I have several comments and queries which should be addressed by the authors in clarifying the methodology ad results of the study:

Response: Thank you.

2. Ethics statement: an ethics statement is lacking in the manuscript

Response: Our original manuscript included the statement, “Institutional ethics approval was obtained from the University of Alberta Research Ethics Board”. We have now added a separate sub-section in the Methods, titled “Institutional Ethics Approval”, where we have elaborated on the ethics approval and implied consent method used in our study. The new sub-section is as follows:

“Institutional Ethics Approval

Prior to beginning the study, we sought and received ethical approval for both the quantitative and qualitative components from the University of Alberta Research Ethics Board (# Pro00058629). Parents and/or caregivers were eligible to participate if they had a child aged 0-17 years who had experienced one or more episodes of acute asthma, bronchiolitis, croup, influenza, strep throat/tonsillitis, pneumonia, sinusitis, and/or wheezing. Participant consent was implied through overt action by completing the survey or publishing public responses online. Prior to participation, we provided interested potential participants with an information letter that described the study and explained consent via overt action. Participants were free to withdraw, end, or modify their participation in the study at any time without consequence, and we retained any data collected only with their permission.”

3. Survey development: The description of the survey development lacks specification necessary for understanding the process and choices made:

a. What process did the authors undergo to reach consensus for the categories

Response: We did not employ any consensus methods to decide on the ten categories. We gathered independent feedback from 10 clinicians/researchers and 8 parents with regard to the draft of the survey. We analysed the written feedback qualitatively, and based on the content analysis, decided on the final list of 10 categories. We have edited the Methods section to include these details: “The first draft of the survey was informed by previous research on the outcomes that are important to clinicians and families of children with asthma [19] and by outcomes frequently reported in the literature. To determine the frequency of outcomes reported in the literature, we identified systematic reviews (SRs) published up to 2013 from the Cochrane Database of Systematic Reviews (n=35) and their included studies. We grouped frequently reported outcomes and those previously identified as important [7, 19, 20] into seven broad categories: disease activity; disease complications; adverse effects of therapy; functional status; social and family outcomes, including quality of life; long-term effects of interventions; and resource utilization.

We sent a draft of our survey to a group of 10 clinicians/researchers and 8 parents who reviewed it independently and provided feedback on its content and comprehensibility. We analysed the written feedback qualitatively. Based on the content analysis, our research team revised the original categories, finally deciding on ten categories of outcomes for the survey. These included outcomes that were identified as important by clinicians and parents that were missing from the draft. The ten categories on the final version of the survey included: major complications; symptoms; length of stay in the emergency department or hospital; needing to see a doctor; returns visits to a doctor or the hospital; reactions to medications; medical test results; maintenance of day-to-day activities; minor complications; and cost of illness. The survey is available as a supplementary file (Supplementary File 1).”

b. I do not understand the how the transition was made from the initial seven categories to the final ten categories. The content of the seven vs. ten categories seems quite different. Some of the ten categories seem to derive from the in initial list (e.g. from disease complications to major and minor complications), but for other categories (e.g. length of stay, needing to see a doctor) I found it difficult to see the transition.

Response: Thank you for this comment. We have clarified the methods for the development of the survey (see above).

c. What and how many content experts and consumers were involved in the pilot testing of the categories (similar as the survey testing?) and which three outcomes were added?

Response: We had 10 clinicians/researchers and 8 parents involved in providing feedback on the drafted survey. We did not in fact add three new categories. Instead, based on content analysis of the feedback we received, we revised the categories such that we ended up with 10 final categories as opposed to the original seven. We have elaborated on our survey development in the Methods section (see above).

4. Final version of the survey: Please provide more insight in the final version of the survey, including presentation as supplementary file. If I understand it well, the ten categories contained individual underlying outcomes? How many outcomes per category were presented. How long did it take to complete the survey?

Response: For improved transparency, we have added a supplementary file (Supplementary File 1) which contains the final version of the survey. Details with regard to the traffic to the survey and its usability (e.g., time take to complete it) are available in our process evaluation, which has been published. Within the present manuscript, we present only the outcomes of the survey and focus groups, as to avoid replicating the data found in our published process evaluation. We have made this more explicit at the beginning of the results section.

5. Outcomes identified: I am confused about the low ranking of ‘maintenance of day-to-day’ activities. This seems in contract with the high ranking of long-term disability under the category major complications. Long-term disability typically refers to limitation in activities and social participation. Please clarify this (apparent) discrepancy. Could it be that the parents did not fully understand these concepts? Especially, since the parents mentioned impact on daily activities in the qualitative assessment via the Facebook sessions. Or am I mixing limitations of the children vs. their parents in daily activities?

Response: Thank you for identifying this apparent discrepancy. Within the first paragraph of our discussion, we have elaborated as to why parents may have ranked day-to-day complications as relatively low priority: “The maintenance of day-to-day activities was of relatively low importance to parents, seemingly contradicting the highly ranked importance of major complications and long-term disability. A previous study of asthma outcomes found that parents were more concerned about the long-term compared to short-term beneficial and harmful effects of therapy [19]. Because we did not quantify the temporality of the outcomes on the survey or in the discussions, we can only presume that parents interpreted the maintenance of day-to-day activities as a short-term outcome resulting from relatively minor illness. Although major complications can result in long-term or permanent changes to daily routines, they could also be potentially life threatening. When presented with the possibility of serious complications that could limit long-term quality of life, the maintenance of one’s daily routine may seem relatively unimportant.”

6. Related to the comment above: did the questions and discussions distinguish in short-term vs. long term outcomes? I would argue that parents are concerned about acute symptoms on the short term, while long-term outcomes would be more related to daily activities.

Response: These are our thoughts exactly. We did not provide any details to parents as to the temporality of the outcomes on the list (e.g., short-term vs. long-term). This may have affected how the parents prioritized the outcomes. In addition to the paragraph added to the discussion (see above, in response to #5), we have added the following to our Limitations: “As we did not provide any details to parents as to the temporality of the outcomes on our survey and in our discussions, we were not able to determine whether short- or long-term complications were more important to parents.”

7. Outcomes reported in the literature: I do not understand what 29 ‘unique’ outcomes means, derived from the 221 outcomes identified. Do you mean grouping of the 221 outcomes in 29 clusters of outcomes?

Response: Out of the total 221 outcomes extracted from the Cochrane systematic reviews, many of these were identical (i.e., duplicates). Out of all the outcomes that were reported, 29 individual outcomes were isolated. We have clarified this point in the results section: “We extracted 221 outcomes from the 35 Cochrane SRs; the same outcomes were often reported in more than one SR. Out of the full list of reported outcomes, we isolated 29 individual outcomes each of which was reported in 1 to 26 of the SRs.”

8. Facebook discussions: An interesting finding to me was that the Facebook discussions revealed other themes more related to the process of care (interaction with health care providers, illness education) than outcomes of care. I suggest reflecting on these differences in the Discussion section.

Response: Thank you for drawing our attention to this detail. We have elaborated on this finding in the first paragraph of our discussion: “Not surprisingly, parents who participated in the focus groups were also concerned about process measures, like wait times, communication with healthcare providers, and managing their child’s care at home. Although parents did not explicitly make the link, research has indicated that certain care processes, e.g., patient-centeredness, may contribute to better health outcomes [29]. For example, family centered care is associated with improved clinical outcomes for children and greater satisfaction with care [30]. The approach is characterized by honest communication between families and healthcare providers; policies and procedures that are tailored to the needs of families and children; ensuring support for families and children; and empowering them to participate in care decisions [30].”

9. Only four parents contributed to the Facebook discussions. The authors should even stronger emphasize this limitation.

Response: We agree. We have elaborated on this challenge within the Limitations: "We had great difficulty engaging parents in the qualitative discussion and only elucidated responses from four participants. This seriously limited our ability to make informed inferences with regard to parents' quantitative ranking of the outcomes, leaving these mainly open to interpretation. For this reason, our understanding of the reasoning behind parents' ranking of the outcomes, and the content of the emergent qualitative themes are only preliminary. Further work is required to develop a more comprehensive understanding of why some ARI-related outcomes are more important to parents than others."

10. I suggest further reflection on comparing outcomes identified via the parents vs. the literature. Can they really be compared?

Response: Thank you for pointing this out. We have added a discussion paragraph as follows: "Good practice in clinical trials includes selecting a primary outcome that measures a clinically relevant and important treatment benefit [37]. Likewise, we found that the bulk of the research in child health focused on biological outcomes, with relatively little attention being paid the psychosocial impact of illness [20, 38, 39]. Still, there remains room in research for the measurement of outcomes important to patients and their families. Standards for Research in (StaR) Child Health, founded in 2009, brings together clinical and methodological experts to develop and promote the uptake of evidence-based guidance for child health research [40]. They assert that trialists should measure the effects of interventions more comprehensively; by measuring long-term outcomes and those that are relevant to decision-makers and families, the findings of trials will be of greater value [20]."

Reviewer 2

1. This is an interesting and well designed study.

Response: Thank you.

2. In table 1, one of the ARI conditions is "Strep throat/tonsillitis" whereas in the Methods section it is described as pharyngitis/tonsillitis so the study authors could consider choosing one of the two for consistency.

Response: Thank you for noticing this. We have chosen "strep throat/tonsillitis" and have ensured that we are consistent throughout the manuscript.

3. The categorized items are shown as ranked by the parents in table 2, but there is no description of how the ranking calculation is done - could this be clarified? i.e. it would be useful to know how many parents thought the different categories were important, since there is a difference between a 1st ranked outcome that was ranked by 100/110 parents vs a 10th ranked outcome that might only be ranked by 1/110 parents.

Response: Thank you for pointing this out. We have clarified our methods as follows: "Survey data were analyzed using SPSS (v. 22.0, IBM Corporation, Armonk, NY) and described using descriptive statistics (mean±SD; rank order). To determine the rank order of the outcomes most important to parents, we allocated each outcome 10 points when it was chosen as a top concern, and 8, 6, 4, and 2 points if it was chosen as the second, third, fourth, or fifth most important concern, respectively.

After tallying the points for each of the 10 outcomes, we ordered these from largest to smallest to develop the ranked priority list.”

4. In addition, the outcomes selected for the survey are described as being obtained from Cochrane systematic reviews, but it would be helpful to know how these were selected - were all outcomes described in the SRs retained? Did some of the study authors select which outcomes were to be retained in collaboration with the individuals involved in the pilot testing portion?

Response: We retained all of the outcomes, and many of these were reported in more than one Cochrane SR. We have elaborated on our methods as follows: “To compare the published literature to the patient-important outcomes identified by parents, we collated a list of all of the outcomes, and calculated the number of SRS in which each outcome had been reported. We then grouped all of the outcomes into percentile ranges based on the number of SRs in which each was reported (‘frequently reported’: 75th-99th percentile; ‘moderately reported’: 25th-74.9th percentile; and ‘infrequently reported’: <25th percentile).”

VERSION 2 – REVIEW

REVIEWER	Philip van der Wees Radboud university medical center
REVIEW RETURNED	19-Oct-2017

GENERAL COMMENTS	The authors have addressed my concerns adequately and the novel approach of the study is of added value for building knowledge in identifying patient relevant outcomes.
--

REVIEWER	Mikael Lavigne North York General Hospital, Toronto, Ontario, Canada
REVIEW RETURNED	07-Nov-2017

GENERAL COMMENTS	Thank you for addressing the questions and requested modifications. The changes made address the majority of the concerns laid out by the reviewers. I would recommend two further minor changes. 1) When stating that "We wrote the materials at a sixth grade reading level", it would be helpful to know what software or method you used to assist others who might wish to pursue this process. 2) On page 7, a "Digital sliding scale" is described. I believe that this digital sliding scale produces a score of 1 to 100, as the parent ranking of importance of outcomes is a mean score out of 100, but this is not stated in the text. This would be helpful to clarify so the reader understands how the parent ranking scores are generated.
--

VERSION 2 – AUTHOR RESPONSE

Reviewer 1

1. The authors have addressed my concerns adequately and the novel approach of the study is of added value for building knowledge in identifying patient relevant outcomes.

Response: Thank you.

Reviewer 2

1. Thank you for addressing the questions and requested modifications. The changes made address the majority of the concerns laid out by the reviewers.

Response: Thank you.

I would recommend two further minor changes.

2. When stating that "We wrote the materials at a sixth grade reading level", it would be helpful to know what software or method you used to assist others who might wish to pursue this process.

Response: To ascertain the reading level of our materials, we requested readability statistics from the proofing options in Microsoft Office Word. When the readability statistics option is turned on, Word returns the Flesh-Kincaid Grade Level of the document, or of the highlighted text following proofing for spelling and grammar. We have added these details to the manuscript.

3. On page 7, a "Digital sliding scale" is described. I believe that this digital sliding scale produces a score of 1 to 100, as the parent ranking of importance of outcomes is a mean score out of 100, but this is not stated in the text. This would be helpful to clarify so the reader understands how the parent ranking scores are generated.

Response: We have clarified the use of the digital sliding scale in our methods, as follows:

Within "Data Collection": "Then, individual outcomes were presented and parents were asked to indicate their importance using a digital sliding scale, conceptually similar to a visual analog scale. The sliding scale asked parents to rank the importance of each outcome by providing it with a score ranging from from 1 (not important [/concerning] at all) to 100 (extremely important [/concerning])."

Within "Quantitative Analysis": "We then grouped all of the outcomes into percentile ranges based on the number of SRs in which each was reported ('frequently reported': 75th-99th percentile; 'moderately reported': 25th-74.9th percentile; and 'infrequently reported': <25th percentile). Similarly, we calculated the mean score given to each outcome by parents on the digital sliding scale (i.e., scores from 1 to 100), and grouped the outcomes into percentile ranges ('most important': 75th-99th percentile; 'moderately important': 25th-74.9th percentile; and 'least important': <25th percentile)."